# Anti-Obesity Effect of Extract from *Nelumbo Nucifera* L., *Morus Alba* L., and *Raphanus Sativus* Mixture in 3T3-L1 Adipocytes and C57BL/6J Obese Mice

**DOI:** 10.3390/foods8050170

**Published:** 2019-05-19

**Authors:** Wan-Sup Sim, Sun-Il Choi, Bong-Yeon Cho, Seung-Hyun Choi, Xionggao Han, Hyun-Duk Cho, Seung-Hyung Kim, Boo-Yong Lee, Il-Jun Kang, Ju-Hyun Cho, Ok-Hwan Lee

**Affiliations:** 1Department of food science and biotechnology, Kangwon National University, Chuncheon 24341, Korea; simws9197@naver.com (W.-S.S.); docgotack89@hanmail.net (S.-I.C.); bongyeon.cho92@gmail.com (B.-Y.C.); zzaoszz@naver.com (S.-H.C.); xionggao414@hotmail.com (X.H.); 2Haram Co. Ltd. Jeungpyeong 27914, Korea; hansol305@naver.com; 3Institute of Traditional Medicine and Bioscience, Daejeon University, Daejeon 34520, Korea; sksh518@dju.kr; 4Department of Food Science and Biotechnology, CHA University, Seongam, Gyeonggi 13488, Korea; bylee@cha.ac.kr; 5Department of Food Science and Nutrition, Hallym University, Chuncheon 24252, Korea; ijkang@hallym.ac.kr

**Keywords:** anti-obesity, *Nelumbo nucifera* L., *Morus alba* L., *Raphanus sativus*, 3T3-L1 adipocyte, C57BL/6J mice

## Abstract

The antioxidant and anti-adipogenic activities of a mixture of *Nelumbo nucifera* L., *Morus alba* L., and *Raphanus sativus* were investigated and their anti-obesity activities were established in vitro and *in vivo*. Among the 26 different mixtures of extraction solvent and mixture ratios, ethanol extract mixture no. 1 (EM01) showed the highest antioxidant (α,α-Diphenyl-β-picrylhydrazyl, total phenolic contents) and anti-adipogenic (Oil-Red O staining) activities. EM01 inhibited lipid accumulation in 3T3-L1 adipocytes compared to quercetin-3-O-glucuronide. Furthermore, body, liver, and adipose tissue weights decreased in the high-fat diet (HFD)-EM01 group compared to in the high-fat diet control group (HFD-CTL). EM01 lowered blood glucose levels elevated by the HFD. Lipid profiles were improved following EM01 treatment. Serum adiponectin significantly increased, while leptin, insulin growth factor-1, non-esterified fatty acid, and glucose significantly decreased in the HFD-EM01 group. Adipogenesis and lipogenesis-related genes were suppressed, while fat oxidation-related genes increased following EM01 administration. Thus, EM01 may be a natural anti-obesity agent.

## 1. Introduction

Obesity is defined as excessive weight gain, particularly inordinate fat accumulation in the body [1]. Due to rapid economic growth and westernized diets, the number of patients with obesity has increased [2], resulting in increased rates of diseases such as type 2 diabetes [3], hypertension [4], hypercholesterolemia [5], and hyperlipidemia [6].

Obesity occurs because of a combination of overeating, lack of exercise, and neurotransmitters, drugs, and genetic factors [7]. To improve obesity, pharmacological treatment has been applied. However, because these treatments involve diverse side effects such as diarrhea and vomiting, the development of anti-obesity materials from natural products with few side effects is required [8,9].

*Nelumbo nucifera* L contains several flavonoids and alkaloids and has recently been used as a plain or blended tea to treat obesity in China [10]. *Morus alba* L is abundant in polyphenols such as rutin and quercetin and has been used to treat dyslipidemia, diabetes, fatty liver disease, and hypertension [11]. *Raphanus sativus* is an annual herb belonging to the cruciferous family and has been reported to be effective for treating hyperlipidemia by decreasing blood sugar level [12]. Some researchers evaluated the anti-obesity effects of *Eriobotrya japonica* and *N. nucifera* in adipocytes and obese mice using a single material or mixture, but few studies have examined mixtures of three natural materials [13]. Because each material has a different physiological activity, it is necessary to confirm the physiological activities of each according to the type and mixing ratio. These materials may exert synergistic effects, but there might be also negative effects [14,15].

In this study, we blended *N. nucifera* L., *M. alba* L., and *R. sativus* in different mixing ratios and selected the optimal mixed material based on antioxidant and anti-adipogenic experiments. The anti-obesity effects of optimal mixed material were evaluated in 3T3-L1 adipocytes and C57BL/6J obese mice.

## 2. Materials and Methods

### 2.1. Sample Preparation

*Nelumbo nucifera* leaves, *M. alba* leaves, and Dried *R. sativus* root were supplied by Haram Co. Ltd. (Jeungpyeong, Korea) and freeze-dried and mixed in 13 ratios. The mixing ratio of *N. nucifera* L*, M. alba* L, *R. sativus* is 80:20:0 (M1), 70:30:0 (M2), 60:40:0 (M3), 50:50:0 (M4), 80:0:20 (M5), 70:0:30 (M6), 70:20:10 (M7), 60:30:10 (M8), 60:20:20 (M9), 50:30:20 (M10), 100:0:0 (M11), 0:100:0 (M12), 0:0:100 (M13). These materials were extracted by boiling and reflux with hot water (WM) and 70% ethanol (EM) at 60 °C for 12 h. Then they were filtered, concentrated by a vacuum evaporator (Rotavapor R-200; Buchi, Flawil, Switzerland), and freeze-dried (Bondiro; Il Shin Lab Co. Ltd., Seoul, Korea) to obtain a powder [16].

### 2.2. Antioxidant Activity Analysis

We performed a DPPH (α,α-Diphenyl-β-picrylhydrazyl) assay as described previously with some modifications [17]. First, 0.8 mL of 0.4 mM DPPH solution was added to 0.2 mL of the sample, and the mixture was incubated in the dark for 10 min. Next, it was measured at the 517 nm absorbance using a microplate reader (Molecular Devices, Sunnyvale, CA, USA).

DPPH radical scavenging activity (%) = (1−(A_experiment_/A_control_)) × 100(1)

A_experiment_: Absorbance of experimental group, A_control_: Absorbance of control group.

The total phenolic contents were determined using Folin-Ciocalteu’s colorimetric method [18]. First, 1 mL of sample was added to 10% Folin-Ciocalteu reagent and 2% Na_2_CO_3_ reagent in order and incubated for 1 h in the dark. It was measured at the 750 nm absorbance using a microplate reader. Gallic acid was used as a standard and the total phenolic contents were calculated from the standard calibration curve (*y* = 19.12*x* − 0.0261, *R*^2^ = 0.9992).

### 2.3. Cell Culture and Differentiation

3T3-L1 preadipocytes were obtained from American Type Culture Collection (CL-173, ATCC, Manassas, VA, USA) and grown in culture plates containing Dulbecco’s modified Eagle’s medium (Lonza, Basel, Switzerland) supplemented with 10% bovine serum (Gibco, Grand Island, NY, USA) and 1% penicillin/streptomycin (P/S; Gibco) kept at 37 °C and 5% CO_2_ incubator condition. 3T3-L1 preadipocytes were differentiated to adipocytes at the 2 day after confluence by exchanging with MDI medium (DMEM containing 10% fetal bovine serum (Gibco), 1% P/S, 0.5 mM 3-isobutyl-1-methylxanthine (Sigma-Aldrich, Saint Louis, MO, USA), 1 μM dexamethasone (Sigma-Aldrich), and 1 μg/mL insulin (Gibco)). Every 2 days during incubation, the culture medium was changed to DMEM containing 10% fetal bovine serum and 1 μg/mL insulin with extracts until day 6 [19].

### 2.4. Cell Viability Assay

3T3-L1 preadipocytes were plated into 96-well plates (1 × 10^6^ cells/well) and differentiated with MDI medium along with 100 μg/mL extracts since the 2 day after confluence for 6 days. At the day 6, Mixed XTT (2,3-bis(2-methoxy-4-nitro-5- sulfophenyl)-2*H*-tetrazolium-5-carboxanilide) and PMS (*N*-methyl dibenzopyrazine methyl sulfate reagents) (WelGene, Seoul, Korea) were included into the medium and incubated for 4 h at 37 °C. Then the soluble formazan salt generated in the medium was measured at 450 nm against 690 nm using a microplate reader [20].

### 2.5. Oil-Red O Staining Assay

The amount of lipid accumulation of 3T3-L1 cells differentiated on a 24-well plate for 6 days was determined using Oil-Red O staining. In brief, the cells were washed with phosphate-buffered saline (PBS; Gibco) and fixed in 10% formaldehyde in distilled water for 1 h. Next, the cells were dried with 60% isopropanol and stained with Oil-Red O (Sigma-Aldrich) solution for 1 h, and then washed with distilled water. The stained lipids were eluted with 100% isopropanol and measured using a microplate reader at 490 nm.

### 2.6. Animal Experiment Design

Eight-week-old male C57BL/6J mice were obtained from Korea BioLink (Eumseong, Korea) and maintained under standard light (12:12-h light/dark cycle), temperature (22 ± 2 °C) and humidity (55 ± 15%) conditions. The diets included a normal diet (AIN-76A; Research Diets, Inc., New Brunswick, NJ, USA) and high-fat diet (HFD; D12492; Research Diets, Inc.), and obesity was induced in the mice fed an HFD for 2 weeks. After 2 weeks, the mice were randomly divided into eight groups; C57BL/6J normal group, 60% kcal HFD control group, positive control group (*Garcinia cambogia*, GC_245 mg/kg), EM11_100 mg/kg group, EM12_100 mg/kg group, EM01_50 mg/kg group, EM01_100 mg/kg group, quercetin-3-O-glucuronide (Q3OG) 10 mg/kg group; *n* = 10). Meanwhile, *Garcinia cambogia* is an ingredient in dietary supplements for weight loss [19]. These test substances were taken orally with saline solution once a day for 8 weeks. The mice had free access to food and water, and their body weight and calorie intake were measured weekly. Biochemical measurements were analyzed after treatment of the candidate materials for 8 weeks and glucose tolerance test were performed 3 days later. Experiments were carried out with the approval of the Institutional Animal Care and Use Committee of Kangwon University for the ethical and scientific feasibility study and effective management of animal experiments (Permit No. KIACUC-12-0140).

### 2.7. Glucose Tolerance Test

After administering the anti-obesity candidate materials for 8 weeks, mice were fasted for 12 h and the blood glucose level was measured at 0, 15, 30, 45, 60, and 75 min after glucose (1 g/kg) peritoneal administration. Blood for glucose measurement was obtained from the tail vein of mice and measured using a serum analyzer (Accutrend plus GCTL Cobas Roche, Basel, Switzerland).

### 2.8. Serum Biochemical Parameter Analysis

After administering the anti-obesity candidate materials for 8 weeks, blood samples were collected by cardiac puncture using centrifugation (2000× *g*, 4 °C, 15 min) and stored at −74 °C. Alanine aminotransferase (ALT), aspartate aminotransferase (AST), and creatinine, which are indices of liver and kidney function, and total cholesterol (TC), high-density lipoprotein cholesterol (HDL-cholesterol), low-density lipoprotein cholesterol (LDL-cholesterol), triglyceride (TG), non-esterified fatty acid (NEFA), and glucose, which are indices of lipid content, were measured using a biochemical automatic analyzer (Hitachi-720, Hitachi Medical, Tokyo, Japan).

### 2.9. Serum Adipokine Analysis

Also, adiponectin, leptin, and insulin-like growth factor-1 (IGF-1) were isolated in blood samples collected by cardiac puncture after administration of anti-obesity candidate materials for 8 weeks. Each antibody was diluted in coating buffer and coated on a microwell for overnight incubation at 4 °C. Each well was washed three times with washing buffer, and 100 μL of serum was dispensed, incubated at room temperature for 1 h, and washed twice with washing buffer. Thereafter, 100 μL of the antibody avidin-horseradish peroxidase conjugate was added and incubated at room temperature for 1 h, and then washed again. The TMB substrate was dispensed in 100 μL, incubated in the dark for 30 min, and treated with 50 μL of stop solution, after which absorbance was measured at 450 nm.

### 2.10. Real-Time Polymerase Chain Reaction (RT-PCR)

Total cellular RNA was extracted from the liver, epididymal, and abdominal subcutaneous adipose tissue using a homogenizer and Trizol reagent (Sigma-Aldrich). Total RNA was used for cDNA synthesis with the One-step SYBR Green PCR kit (AB Science, Avenue George V, France). The Applied Biosystems 7500 Real-Time PCR system (Applied Biosystems, Foster City, CA, USA) used for real-time quantitative PCR. The probes containing the fluorescence reporter dye 6-carboxy-fluorescein (Applied Biosystems) was used to indicate mRNA gene expression. The mouse glyceraldehyde-3-phosphate dehydrogenase probe (VIC/MGB probe, primer limited, 4352339E, Applied Biosystems) was used as an internal standard. The sequence of the used primer was as follows; Leptin sense (5′-AACCCTTACTGAACTCAGATTGTTAG-3′) and antisense (5′-TAAGTCAGTTTAAATGCTTAGGG-3′); PPARγ sense (5′-ATGCCATTCTGGCCCACCAACTT-3′) and antisense (5′-CCCTTGCATCCTTCACAAGCATG-3′); PPARα sense (5′-GCCTGTCTGTCGGATGT-3′) and antisense (5′-GGCTTCGTGGATTCTCTTG-3′); Adiponectin sense (5′-TTCAAATGAGATTGTGGGAAAAT-3′) and antisense (5′-ACCGATACAGTACAGTACAGTA-3′); UCP1 sense (5′-CGACTCAGTCCAAGAGTACTTCTCTTC-3′) and antisense (5′-GCCGGCTGAGATCTTGTTTC-3′); UCP2 sense (5′-TTCAAATGAGATTGTGGGAAAAT-3′) and antisense (5′-ACCGATACAGTACAGTACAGTA-3′); ACX1 sense (5′-CAGGAAGAGCAAGGAAGTGG-3′) and antisense (5′-CCTTTCTGGCTGATCCCATA-3′); DGAT1 sense (5′-TGCTACGACGAGTTCTTGAG-3′) and antisense (5′-CTCTGCCACAGCATTGAGAC-3′); SCD1 sense (5′-CATCGCCTGCTCTACCCTTT-3′) and antisense (5′-GAACTGCGCTTGGAAACCTG-3′); SREBP1c/ADD1 sense (5′-AGCCTGGCCATCTGTGAGAA-3′) and antisense (5′-CAGACTGGTACGGGCCACAA-3′); ACS1 sense (5′-TCCTACAAAGAGGTGGCAGAACT-3′) and antisense (5′-GGCTTGAACCCCTTCTGGAT-3′); CPT1b sense (5′-GTCGCTTCTTCAAGGTCTGG-3′) and antisense (5′-AAGAAAGCAGCACGTTCGAT-3′); FAS sense (5′-CTGAGATCCCAGCACTTCTTGA-3′) and antisense (5′-GCCTCCGAAGCCAAATGAG-3′); GAPDH VIC (5′-TGCATCCTGCACCACCAACTGCTTAG-3′). The PCR conditions were 50 °C for 2 min, 94 °C for 10 min, 95 °C for 15 s, and 45 °C for 1 min of 40 cycles.

### 2.11. Statistical Analysis

The results are expressed as the mean ± standard deviation of triplicate experiments. An analysis of variance and Duncan’s multiple range tests were used to determine the significance at the *p* < 0.05 level.

## 3. Results

### 3.1. Effects of 26 Extracts by Mixture Ratio of N. Nucifera L., M. alba L., R. Sativus on the Antioxidant and Anti-Adipogenic Activities

α,α-Diphenyl-β-picrylhydrazyl (DPPH) is a very stable free radical and representative reaction substance used to measure antioxidant capacity. DPPH radical scavenging activity assay is a method that utilizes the principle that a purple compound is discolored as yellow when radicals are eliminated through hydrogen donation in a phenolic compound or flavonoid with a hydroxyl radical (-OH) [21]. For the extraction method, the radical scavenging activities of ethanol extracts were superior to those of hot water extracts. For the mixing ratio, EM05 (84.30%), EM06 (83.00%), and EM01 (81.65%) were superior in order, but there was no significant difference between these samples (Figure 1A). 

The content of phenol, a representative antioxidant, was highest in EM03 (49.00 mg GAE/g), EM02 (48.40 mg GAE/g), and EM01 (47.90 mg GAE/g), with no significant difference between these samples (Figure 1B).

No cytotoxicity or changes in morphology were observed at the concentration of 100 μg/mL mixtures by XTT assay (Figure 1C). Next, the anti-adipogenic activity was measured by Oil-Red O staining, and EM01 (75.30%) was found to be the most effective mixture for inhibiting lipid accumulation among the 26 mixtures (Figure 1D). Therefore, we selected EM01 as an optimal mixture and carried out anti-obesity experiments using in vitro and in vivo models.

### 3.2. Effect of EM01 on Lipid Accumulation

We previously found that quercetin-3-O-glucuronide is a bioactive compound in mixed materials [16]. In antioxidant and anti-obesity experiments, EM01 (100 μg/mL) was measured to determine its anti-obesity activity with quercetin-3-O-glucuronide (7.8 μM). As shown in Figure 2, EM01 (80.73%) showed better lipid accumulation inhibitory activity than the single materials, EM11 (84.92%), EM12 (86.39%), and single bioactive compound treatment (84.65%). There may have been a synergistic interaction in EM01 between quercetin-3-O-glucuronide and other compounds.

### 3.3. Effects of EM01 on Body Weight, Food Intake, FER, Organ Weight, and Adipose Tissue Weight in HFD-Induced Obese Mice

There was no significant difference in the initial body weight between experimental groups, but the final body weights in different groups were significantly different. The body weight increased significantly in the HFD-CTL group compared to in the ND group, suggesting that obesity was induced by the HFD. The HFD-EM01 group (100 μg/mL) showed a higher weight loss rate than the HFD-CTL group (Figure 3A). Food intake was not significantly different between groups except for the ND group, and the food efficiency ratio (FER) was significantly decreased in the HFD-EM01 group (100 μg/mL) (Figure 3B,C).

The kidney and spleen weights were not significantly different among groups (Figure 3D). Liver weight increased significantly in the HFD-CTL group compared to in the ND group, and weight decreased significantly in the HFD-EM01 and HFD-Q3OG groups compared to in the HFD-CTL group. According to Figure 3E, the weight of the kidney adipose tissue significantly increased in the HFD-CTL group compared to in the ND group, but no significant inhibitory effect was observed in any group. The weight of abdominal subcutaneous fat, epididymal, and intestine adipose tissue increased significantly in the HFD-CTL group compared to in the ND group, but there was an inhibitory effect on fat accumulation in the HFD-EM01 and HFD-Q3OG group compared to the positive control (HFD-GC group).

### 3.4. Effects of EM01 on glucose tolerance in HFD-induced obese mice

When glucose tolerance occurs, blood glucose levels do not rise despite glucose administration, which is common in obese patients [22]. To investigate the effect of EM01 administration on glucose-induced hyperglycemia, glucose was orally administered, and the glucose tolerance test was performed over time. Blood glucose levels in all groups increased after 30 min of glucose injection. After 60 min of glucose injection, blood glucose levels in EM01 and Q3OG administrated groups dropped markedly even close to ND-group. On the other hand, it was confirmed that blood glucose levels did not decrease rapidly in EM11 and EM12 administrated groups (single material) (Figure 4).

### 3.5. Effects of EM01 on the Serum Lipid Profile in HFD-Induced Obese Mice

The HFD-CTL group showed significant increases in all parameters of the serum lipid profile compared to the ND group. The TC level was significantly decreased in the HFD-EM01 and HFD-Q3OG groups compared to in the HFD-CTL group (Figure 5A). HDL-cholesterol was lowered by more than LDL-cholesterol in these groups (Figure 5B,C). TC may decrease when LDL-cholesterol levels, often referred to as bad hormones, are suppressed [23]. The TG level was higher in the HFD-CTL group than in the ND group, but there was no significant difference between all groups compared to HFD-CTL (Figure 5D). AST, ALT, and creatinine are used as liver toxicity marker [24] and their levels were significantly lowered by EM01 administration (Figure 5E,F). This suggests that the 8-week oral administration did not affect liver and kidney toxicity in obese mice.

### 3.6. Effects of EM01 on the Energy Balancing Metabolism in HFD-Induced Obese Mice

Adipokine is a hormone specifically secreted from adipose tissue that affects endocrine system function. We analyzed adiponectin, leptin, IGF-1, which plays an important role in normal growth and health maintenance, NEFA, and glucose, which is associated with energy homeostasis [25]. Adiponectin levels in the serum were significantly decreased in the HFD-CTL group compared to in the ND group, but there was a significant increase in the HFD-EM01 and HFD-Q3OG groups compared to in the HFD-CTL group (Figure 6A). Leptin, IGF-1, NEFA, and glucose levels in the serum were significantly increased in the HFD-CTL group compared to in the ND group, while HFD-EM01 and HFD-Q3OG group were significantly decreased compared to the HFD-CTL group (Figure 6B–E).

### 3.7. Effects of EM01 mRNA Expression Level of Lipid Metabolism-Related Genes in HFD-Induced Obese Mice

We analyzed the mRNA levels of adipogenesis, lipogenesis, and fatty acid oxidation-related genes in the liver, epididymal adipose tissue, and abdominal subcutaneous fat after 8 weeks of EM01 administration (Figure 7).

As shown in Figure 7A, the mRNA levels of FAS, DGAT1, SCD-1, leptin, SREBP1c, PPARγ in the liver were significantly increased in the HFD-CTL group but decreased in the HFD-EM01 and HFD-Q3OG groups. The mRNA levels of COX1, adiponectin, UCP2, and PPARα increased in the HFD-EM01 and HFD-Q3OG groups compared to in the HFD-CTL group.

As shown in Figure 7B, the mRNA levels of the FAS, leptin in the epididymal adipose tissue increased in the HFD-CTL group, but markedly decreased in HFD-EM01 and HFD-Q3OG groups. The mRNA levels of the ACS1, ACOX1, CPT1b, UCP2, adiponectin, PPARα increased in HFD-EM01 and HFD-Q3OG groups compared to those in the HFD-CTL group.

As shown in Figure 7C, the mRNA level of UCP1 in abdominal subcutaneous fat significantly increased in the HFD-EM01 and HFD-Q3OG groups compared to in the HFD-CTL group.

## 4. Discussion

In recent years, many side effects have been reported for drugs used for therapeutic purposes, and numerous clinical studies have examined natural products and natural product-derived compounds [26,27]. In this study, we confirmed the anti-obesity effect of three natural materials. In previous studies based on these materials, *N. nucifera* L was found to be a medicinal plant that is not only useful for treating gastritis, bleeding, diarrhea, hemorrhoids, and enuresis, but also contains various biologically active components such as polyphenolics, flavonoids, and tannic acid [28]. SOD, CAT, GST played as defence means against the reactive oxygen species in biological systems. TBARS are formed as a byproduct of lipid peroxidation. This material was reported to have anti-oxidative activity by increasing the levels of SOD, CAT, GST and decreasing TBARS level in liver [29] and anti-obesity activity in HFD-induced mice or anti-obesity activity for inducing apoptosis in 3T3-L1 adipocytes, but few studies have examined these effects [30]. *Morus alba* L, which is prepared for medicinal use such as treating headache, fever, and cough, shows pharmacological activities, particularly antidiabetic effects on lowering blood sugar [31]. *Raphanus sativus* is an herbaceous plant belonging to the cruciferous family and has been reported to have excellent anti-obesity efficacy [32]. Recent studies have demonstrated the anti-obesity activity of a single material, but the mechanism action of this material in a mixture has not been evaluated.

We selected EM01, which contains a high content of *Nelumbo nucifera* [16], among the 26 mixtures (13 kinds of hot water extracts and 13 ethanol extracts) through the antioxidant and anti-obesity experiments at in vitro model. EM01 treatment has an anti-obesity effect at in vivo model using C57BL/6J obese mice. The weight of HFD-induced mice was effectively lower and the weight of the adipose tissue was significantly decreased compared with the control.

Type 2 diabetes is one of the most common disabilities caused by obesity. Obesity directly affects insulin function, which causes glucose to enter the cell membrane for energy metabolism, resulting in insulin resistance and type 2 diabetes [33]. A glucose tolerance test was conducted to investigate the glucose processing ability. EM01 effectively lowered the blood glucose level. This result was similar to those of the glucose tolerance test using neferine an alkaloid compound extracted from *N. nucifera* seed [34].

EM01 reduced the serum lipid profile (TC, HDL-cholesterol, LDL-cholesterol, TG) of HFD-induced obese mice. It may be useful for preventing dyslipidemia induced by obesity [35]. AST, ALT, and creatinine are toxicity indicators of the liver and kidney [24]. EM01 reduced the levels of these markers in the serum and improved the function of each organ.

Adipocytes not only play a role as energy reservoirs but also regulate endocrine function. Adipokine, a hormone specifically expressed and secreted from adipocytes such as adiponectin and leptin, is involved in fatty acid and glucose metabolism. IGF-1 plays an important role in regulating adipose tissue growth. These adipokines interact closely with each other and regulate the hormonal system in the body [36]. EM01 significantly increased the adiponectin level and reduced leptin and IGF-1 level in serum. According to a previous study [37], glucose and lipid metabolism are regulated by insulin signaling. Glucose and fatty acid enter cells and are metabolized for glycogenesis and β-oxidation, respectively. Based on these results, our study confirms that EM01 improves the process of bringing glucose and fatty acid into the cells, resulting in anti-obesity and anti-diabetic effects.

Adipogenesis is the process of cell differentiation by which pre-adipocytes become adipocytes. PPARγ is a key factor in adipogenic transcription [1]. In this study, it was found that mRNA expression of PPARγ in the liver tissue was reduced by administration of EM01. Also, the expression levels of FAS, DGAT1, SCD-1, leptin, SREBP1c were decreased, while the expression levels of UCP2, PPARα, ACOX1, and adiponectin increased in the EM01, Q3OG treatment group in the liver. These results suggest that EM01 treatment inhibits adipogenesis of 3T3-L1 adipocytes and improves fatty acid oxidation. EM01 regulated lipid metabolism in epididymal adipose tissue and abdominal subcutaneous fat. Our results demonstrate that EM01 has an anti-obesity effect in 3T3-L1 adipocytes and C57BL/6J obese mice.

## 5. Conclusions

In conclusion, we observed the anti-obesity activity of EM01, which is an optimal mixed material containing *N. nucifera* L., *M. alba* L., and *R. sativus* in vitro and in vivo models. EM01 reduced lipid accumulation and weight gain, fat mass, serum lipid concentration, and mRNA expression levels of lipid-metabolism-related genes in HFD-induced obese mice. Therefore, our findings show that EM01 has potential as an effective material for anti-obesity treatment.

## Figures and Tables

**Figure 1 foods-08-00170-f001:**
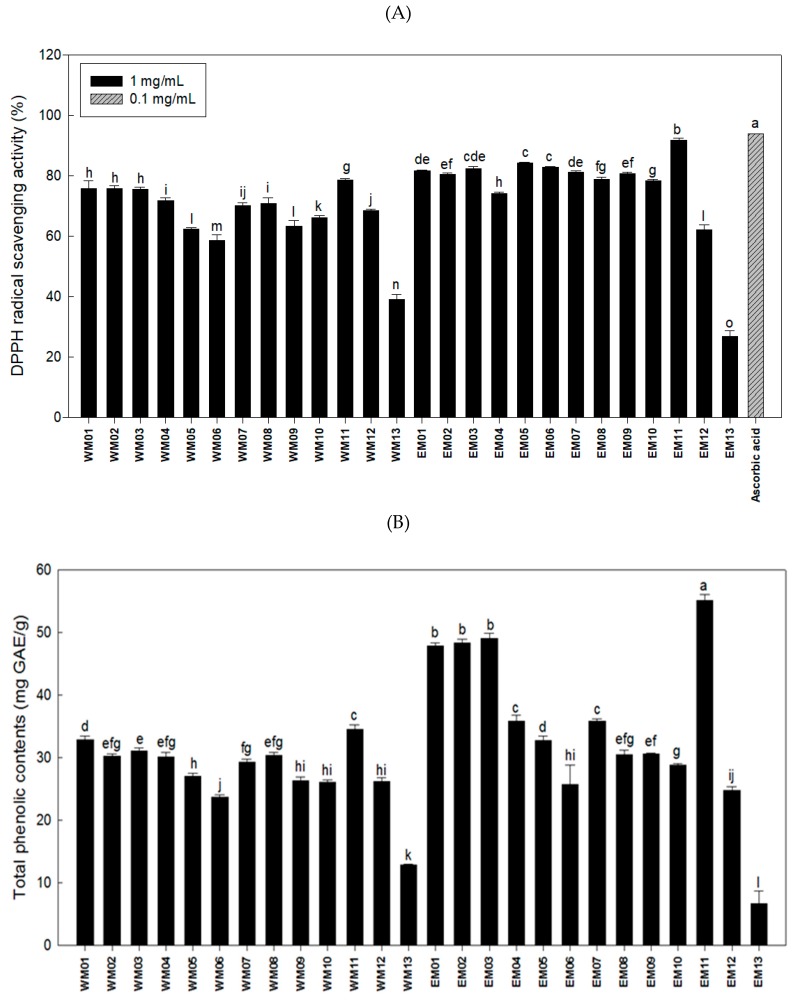
Effects of 26 extracts by mixture ratio of *Nelumbo nucifera* L., *Morus alba* L., and *Raphanus sativus* on antioxidant activity, cell viability, lipid accumulation. (**A**) DPPH radical scavenging activity. (**B**) Total phenolic contents. (**C**) Post-confluent 3T3-L1 preadipocytes were differentiated along with the treatment of each extracts for 6 days. XTT (2,3-bis(2-methoxy-4-nitro-5- sulfophenyl)-2*H*-tetrazolium-5-carboxanilide) and PMS (*N*-methyl dibenzopyrazine methyl sulfate reagents) mixture was added to the medium. After 4 h of incubation, cell viability was found out by calculating the absorbance at 450 nm against 690 nm. (**D**) Stained lipids were extracted and quantified by calculating the absorbance at 490 nm. Data are presented as the mean ± SEM (*n* = 3). Means with different letters on bars indicate that there is a significant difference at *p* < 0.05 by Duncan’s multiple range test.

**Figure 2 foods-08-00170-f002:**
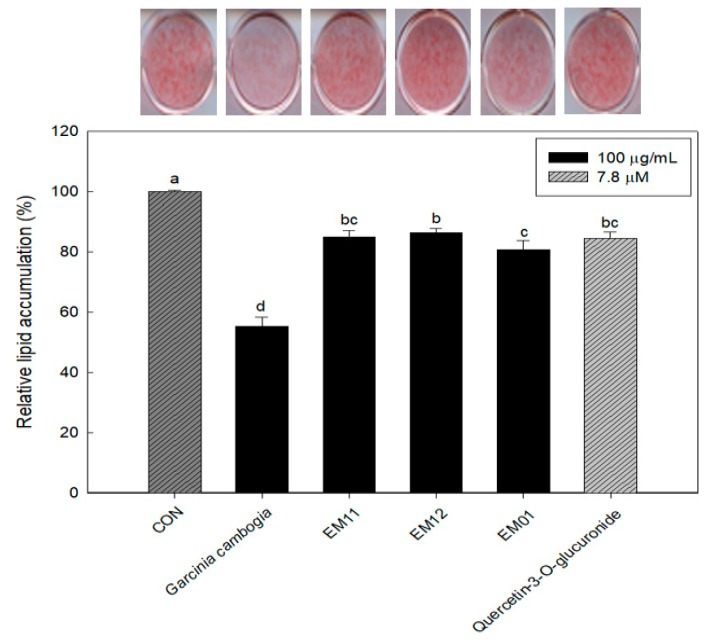
Effect of EM01 and its bioactive compound on lipid accumulation. Post-confluent 3T3-L1 preadipocytes were differentiated along with the treatment of each extracts and its bioactive compound for 6 days. Stained lipids were eluted and quantified by calculating the absorbance at 490 nm. Data are presented as the mean ± SEM (*n* = 3). Means with different letters on bars indicate that there is a significant difference at *p* < 0.05 by Duncan’s multiple range test.

**Figure 3 foods-08-00170-f003:**
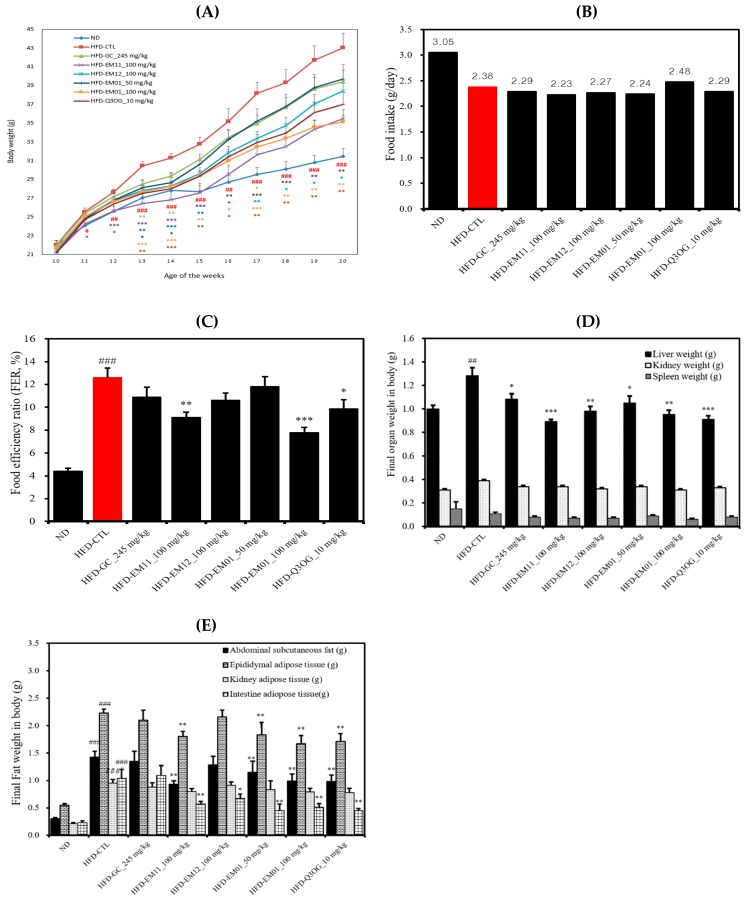
Effects of EM01 on **(A)** body weight, **(B)** food intake, **(C)** FER, **(D)** organ weight, and **(E)** adipose tissue fat weight in high-fat diet (HFD)-induced obese mice. GC, *Garcinia cambogia* extract; HFD, mice were fed a high-fat diet (60% kcal fat); ND, mice were fed a normal diet (10% kcal fat); FER, food efficiency ratio (total weight gain/total food intake × 100). Data are presented as the mean ± SEM (*n* = 12); # *p* < 0.05, ## *p* < 0.01, ### *p* < 0.001 vs. ND; * *p* < 0.05, ** *p* < 0.01 and *** *p* < 0.001 vs. HFD-CTL.

**Figure 4 foods-08-00170-f004:**
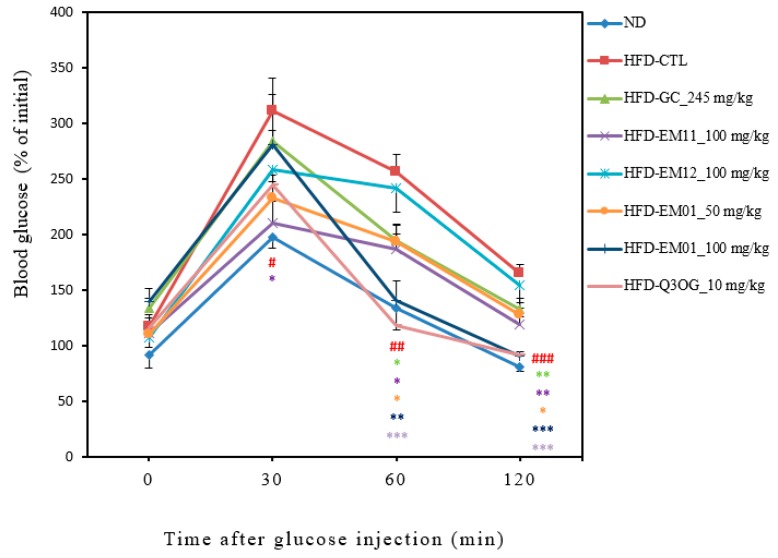
Effects of EM01 on glucose tolerance in HFD-induced obese mice. Data are presented as the mean ± SEM (*n* = 12); # *p* < 0.05, ## *p* < 0.01, ### *p* < 0.001 vs. ND; * *p* < 0.05, ** *p* < 0.01 and *** *p* < 0.001 vs. HFD-CTL.

**Figure 5 foods-08-00170-f005:**
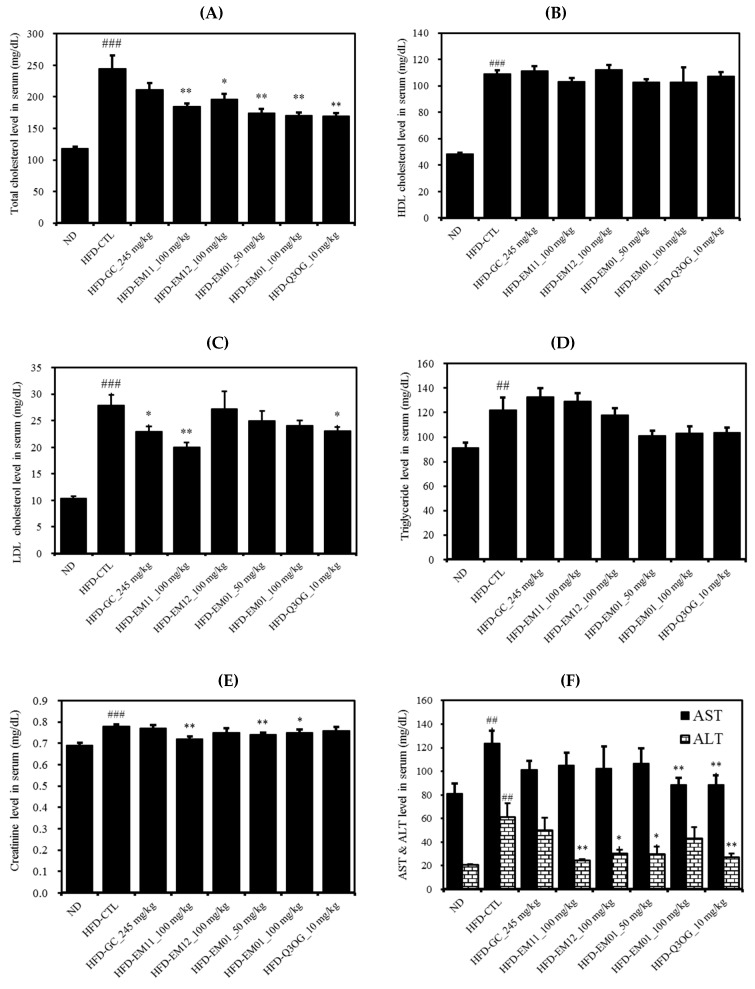
Effects of EM01 on serum lipid profile in HFD-induced obese mice. (**A**) Total cholesterol. (**B**) HDL cholesterol. (**C**) LDL cholesterol. (**D**) Triglyceride. (**E**) Creatinine. (**F**) AST & ALT. Data are presented as the mean ± SEM (*n* = 12); # *p* < 0.05, ## *p* < 0.01, ### *p* < 0.001 vs. ND; * *p* < 0.05, ** *p* < 0.01 and *** *p* < 0.001 vs. HFD-CTL. AST, aspartate aminotransferase; ALT, alanine aminotransferase.

**Figure 6 foods-08-00170-f006:**
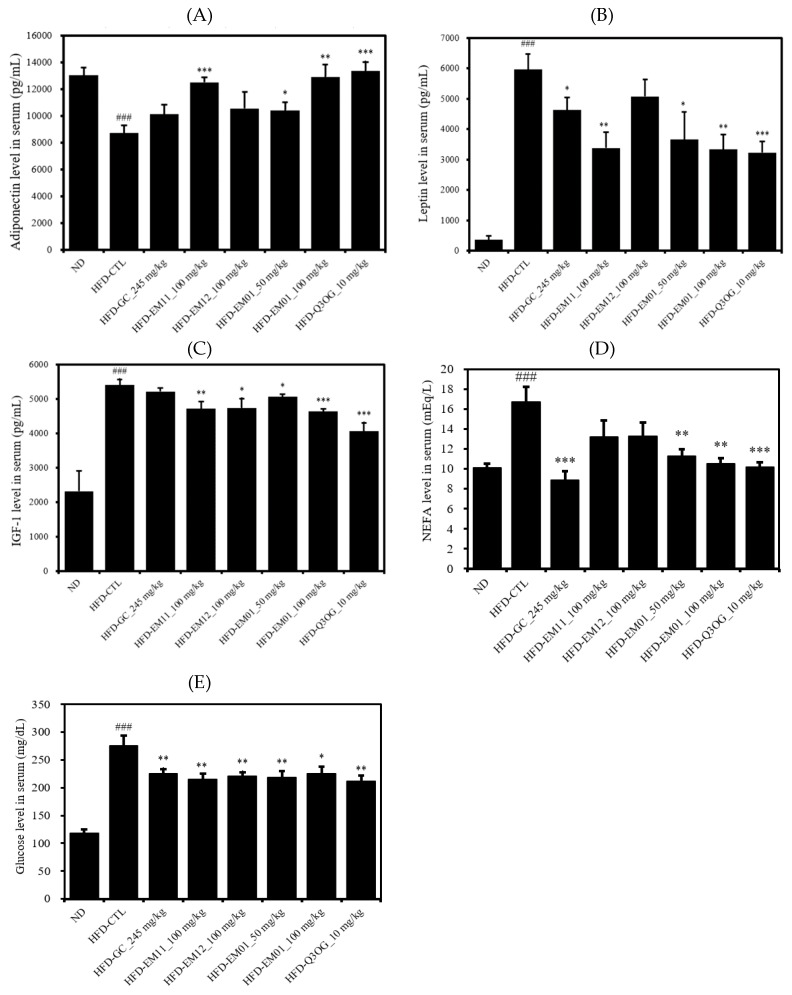
Effects of EM01 on the energy balancing metabolism in HFD-induced obese mice. (**A**) Adiponectin. (**B**) Leptin. (**C**) IGF-1. (**D**) NEFA. (**E**) Glucose. Data are presented as the mean ± SEM (*n* = 12); # *p* < 0.05, ## *p* < 0.01, ### *p* < 0.001 vs. ND; * *p* < 0.05, ** *p* < 0.01 and *** *p* < 0.001 vs. HFD-CTL. IGF-1, insulin-like growth factor-1; NEFA, non-esterified fatty acid.

**Figure 7 foods-08-00170-f007:**
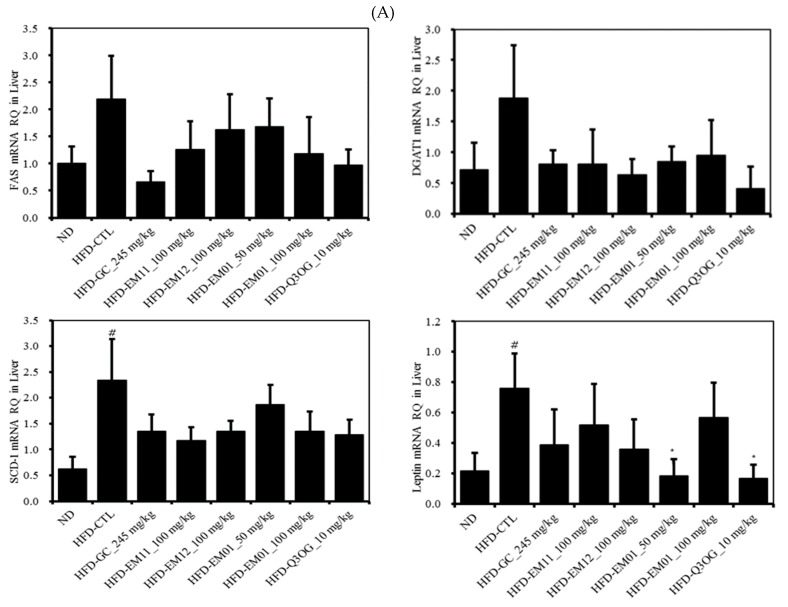
Effects of EM01 on mRNA expression level of lipid metabolism-related genes in HFD-induced obese mice. (**A**) Liver. (**B**) Epididymal adipose tissue. (**C**) Abdominal subcutaneous fat. Data are presented as the mean ± SEM (*n* = 12); # *p* < 0.05, ## *p* < 0.01, ### *p* < 0.001 vs. ND; * *p* < 0.05, ** *p* < 0.01 and *** *p* < 0.001 vs. HFD-CTL. FAS, fatty acid synthase; SCD-1, stearoyl-CoA desaturase-1; SREBP-1c, sterol regulatory element binding protein-1c; PPARγ, peroxisome proliferator-activated receptor γ; DGAT1, diglyceride acyltransferase; UCP, mitochondrial uncoupling proteins; ACOX1, peroxisomal acyl-coenzyme A oxidase 1; PPARα, peroxisome proliferator-activated receptor α; ACS1, acetyl CoA synthetase 1; CPT1b, carnitine palmitoyltransferase 1b.

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
