# Peer review of "Anti-Obesity Effect of Extract from Nelumbo Nucifera L., Morus Alba L., and Raphanus Sativus Mixture in 3T3-L1 Adipocytes and C57BL/6J Obese Mice"

_foods, 2019, doi:10.3390/foods8050170_

Round 1

Reviewer 1 Report

This manuscript written by Wan-Sup Sim and coworkers described the anti-obesity effect of extracts from three plants. In vitro and in vivo assay were performed.

The errors and comments are list below:

1.        line 40, the authors mentioned that westernized diets can cause Obesity. Please provide references.

2.        Line 44, to “improve” obesity….. Did the authors mean reduce?

3.        Line 65, the ratio of M1 did not include R. sativus. I feel the experiment design was very strange. The author emphasized R. sativus in introduction, discussion, and conclusions. However, this plant was not in EM1.

4.        Line 308, 4 Discussion not 5.

5.        Figure 3, the picture comes out blur. Please provide a high resolution picture.

6.        In figure 1, Are there any significant difference between those groups.

Author Response

1.        line 40, the authors mentioned that westernized diets can cause Obesity. Please provide references.

Thank you for this comment. We have added a reference explaining the relationship between westernized diets and obesity.

2.        Line 44, to “improve” obesity….. Did the authors mean reduce?

Thank you for this comment. As you mentioned, the sentence “To improve obesity” means “To reduce the incidence of obesity”. 

3.        Line 65, the ratio of M1 did not include R. sativus. I feel the experiment design was very strange. The author emphasized R. sativus in introduction, discussion, and conclusions. However, this plant was not in EM1.

Thank you for this comment. The purpose of this experiment was to investigate the anti-obesity synergistic effect of extracts from mixed materials compared to single material. If a mixture of two materials is more effective than a mixture of three materials, a mixture of two materials was recommended. As a result, two or three mixed materials were made (For instance, R. sativus was not mixed in EM1, and also M. alba L. was not mixed in M6, M7 mixtures).  

4.        Line 308, 4 Discussion not 5.

Thank you for this comment. We have modified it as you mentioned.

5.        Figure 3, the picture comes out blur. Please provide a high resolution picture.

Thank you for this comment. We have modified Fig 3 with a high resolution picture.

6.        In figure 1, Are there any significant difference between those groups.

Thank you for this comment. As you mentioned, there are significant differences between groups at p < 0.05 by Duncan`s multiple range test. So, Means with different letters on bars were indicated.  

Reviewer 2 Report

Authors described Anti-obesity , antioxidant effect of water and methanolic extract from Nelumbo nucifera L., Morus alba L., and Raphanus sativus mixture in 3T3-L1 adipocytes and C57BL/6J obese mice. Based on several in vitro and in vivo experiments authors selected EM01 combination as most promising.All described experiments were planned and carried out in accordance with applicable standards. However, I would like to point out that the concentration of 1 mg / ml used in the DPPH test is too large and would better characterize true activity at, e.g. 100 μg / ml, and would be comparable to activity on cell cultures. The content of alkaloids should be examined as a significant part responsible for the activity in EMO1.

In sample preparation section there is no information which part of R.sativus was used?  

Author Response

1.        I would like to point out that the concentration of 1 mg / ml used in the DPPH test is too large and would better characterize true activity at, e.g. 100 μg / ml, and would be comparable to activity on cell cultures. 

Thank you for this comment. We agree with that if the sample concentration of the DPPH test was 100 μg/mL, it would be appropriate to compare with the cell experiments. However, we conducted the DPPH test with the aim of identifying significant differences between groups. We will more consider the concentration setting.

2.        The content of alkaloids should be examined as a significant part responsible for the activity in EMO1. 

Thank you for this comment. EM01 consists of N. nucifera L and M. alba L contains various active compounds such as polyphenol, flavonoids. So, we examined the total phenolic contents. As you mentioned, we are totally agreed with your suggestion that the content of alkaloids should be examined as a significant part responsible for the activity in EM01, so we will investigate the content of alkaloids in further study.  

3.        In sample preparation section there is no information which part of R.sativus was used?

Thank you for this comment. Dried R. sativus root was used. We have added it in sample preparation section.
